# Multi-Agent Reinforcement Learning for Online Food Delivery with Location Privacy Preservation

**Suleiman Abahussein** [1] , **Dayong Ye** [1,*] , **Congcong Zhu** [2], **Zishuo Cheng** [3], **Umer Siddique** [4] and **Sheng Shen** [3]

1   Computer Science School, The University of Technology Sydney, Sydney, NSW 2007, Australia; suleiman.abahussein@student.uts.edu.au
2   Faculty of Data Science, City University of Macau, Macau 999078, China; congmaoxian@gmail.com
3   School of Electrical and Information Engineering, The University of Sydney, Camperdown, NSW 2050, Australia; zishuo.cheng@student.uts.edu.au (Z.C.); sheng.shen@sydney.edu.au (S.S.)
4   Department of Computer Science , University of Texas at San Antonio, San Antonio, TX 78249, USA; muhammadumer.siddique@my.utsa.edu
*   Correspondence: dayong.ye@uts.edu.au

**Abstract:** Online food delivery services today are considered an essential service that gets significant attention worldwide. Many companies and individuals are involved in this field as it offers good income and numerous jobs to the community. In this research, we consider the problem of online food delivery services and how we can increase the number of received orders by couriers and thereby increase their income. Multi-agent reinforcement learning (MARL) is employed to guide the couriers to areas with high demand for food delivery requests. A map of the city is divided into small grids, and each grid represents a small area of the city that has different demand for online food delivery orders. The MARL agent trains and learns which grid has the highest demand and then selects it. Thus, couriers can get more food delivery orders and thereby increase long-term income. While increasing the number of received orders is important, protecting customer location is also essential. Therefore, the Protect User Location Method (PULM) is proposed in this research in order to protect customer location information. The PULM injects differential privacy (DP) Laplace noise based on two parameters: city area size and customer frequency of online food delivery orders. We use two datasets—Shenzhen, China, and Iowa, USA—to demonstrate the results of our experiments. The results show an increase in the number of received orders in the Shenzhen and Iowa City datasets. We also show the similarity and data utility of courier trajectories after we use our obfuscation (PULM) method.

**Keywords:** privacy; differential privacy; online food delivery; deep reinforcement learning; multi-agent reinforcement learning



## 1. Introduction

Usage of emerging advanced technologies such as smartphones has grown rapidly and has substantially impacted customer behaviour in online shopping, specifically during the COVID-19 pandemic. Today, online food delivery businesses are considered one of the most widespread businesses worldwide and have grown globally. It is expected that online food delivery will grow to 2.5 bn users by 2027, and it is expected that in the grocery delivery segment, the average revenue per user will be USD 449.00 in 2023 [1]. Today, many people, especially in urban areas, do not have enough time to prepare meals for many reasons such as long working hours; hence, they often turn to online food delivery services that connect restaurants or food outlets with couriers, who then deliver the food to the customer.

Online food delivery applications are considered essential for many people nowadays. However, many of these applications experience many operational issues that reduce their

efficiency. In cities, each area has a different number of food delivery orders compared with other areas. Most food delivery couriers rely on their experience to find areas with high food delivery order demands. However, sometimes they may go to areas with low demand. This issue can reduce the number of orders received by the couriers and lead to a decrease in their income in the long term and can increase the customer waiting time in a busy area as there are not enough couriers, which could reduce customer satisfaction. This is why many companies seek to improve their applications and attempt to increase the number of acquired food delivery orders to help increase the company's and couriers' income and reduce the waiting time for customers in busy areas in order to increase customer satisfaction.

This study introduces a method based on multi-agent reinforcement learning for online food delivery services that utilizes two multi-agent reinforcement learning algorithms. The primary objective of this method is to increase the number of received food delivery orders and increase the long-term income for the couriers. This method also helps to reduce the waiting time for customers in busy areas by guiding couriers to areas with high demand for food orders. A map of the city is split into small grids, each grid represents a small city area, and the agent has to learn to locate the area with high food delivery order demands. This approach enables couriers to find areas with high demand for food delivery orders so that the couriers get more orders, which helps them to increase their long-term income.

While enhancing online food delivery services is important and provides numerous benefits for many people, protecting customer information in online food delivery services is crucial. There are some privacy concerns about leaking sensitive information to users, such as customer location information for online food delivery services. Thousands of food delivery orders are received each day by these platforms, with vast amounts of information collected from the users. These data may be hosted by a third party and may be further processed for training and analysis purposes. Also, the IT department may be operated by a third party. Various access authorities are granted access to different types of data, which allows illegal access to customer data. Furthermore, adversaries can employ various attacks, such as inference attacks, and be able to infer some private information that could pose a serious threat to disclosing customer information, such as the customer's location.

To tackle this issue, a defence method has been proposed to protect the customer's location. The proposed method, called the 'Protect User Location Method' (PULM), aims to maintain the privacy of the customer's location in online food delivery services. This method uses differential privacy (DP) and the Laplace mechanism by injecting Laplace noise into the customer's location and the courier's trajectory. The privacy parameter $\epsilon$ that affects the amount of injected noise depends on two parameters: the city area size and the frequency of customers' online food delivery orders. In small cities, the adversary has a higher opportunity to identify the customer's location due to the small area and fewer number of routes, which makes it easier for the adversary to find the location of the user, whereas it could be more difficult in large cities due to the large geographical area and increased number of roads. Also, when there are a number of food delivery requests from the same customer, the adversary may be able to link the records of the same customer and obtain the customer's private information. Therefore, we inject more noise into the records requested from small cities and for customers with a high number of records of online food delivery orders.

The main contributions of this research are the following:

(i)     We propose a method that aims to improve the efficiency of online food delivery applications by guiding the courier to areas with high demand in order to increase income in the long term.

(ii)    We consider weekdays and weekends as a factor in the agent's learning process to gain better results as the number of orders can vary on weekdays vs. weekends.

(iii)   To show more results and comparisons, this research employs two multi-agent reinforcement learning methods, QMIX and IQL, which aim to increase the number of received orders by couriers and raise long-term income.

(iv)     We use two datasets with different city sizes and different geographic areas to present more results.

(v)      We invent a privacy method called the 'Protect User Location Method' (PULM). This method uses the city area size and customer frequency of online food delivery orders to determine the privacy parameter.

(vi)     We employ differential privacy (DP) by injecting Laplace noise to preserve the customer's location along with the courier's trajectory.

## 2. Literature Review

The section shows the previous work related to the online food delivery service problem and location privacy preservation along with a brief explanation of our solution.

### 2.1. Online Food Delivery Service

In the literature, online food delivery has been broadly studied, yet it remains a challenging problem. Many researchers have tackled the online food delivery problem, and different methods have been proposed. The following are reviews of some previous studies in this field.

Chen et al. [2] focus on same-day delivery by drones and vehicles. By vehicles, multiple packages can be delivered along one route, but the travel is fairly slow. By drones, travel is much faster, but they frequently require battery charging and have limited capacity. Their proposed method is based on a deep Q-learning approach. The method learns to assign a delivery to a new customer via either vehicles or drones or gives the option of not providing the service at all. Liu et al. [3] focus on route planning. In order to capture the preferences of delivery personnel from their historical GPS trajectories and recommend preferred routes, they developed a deep inverse reinforcement learning (IRL) algorithm. Also, the Dijkstra algorithm was adopted in their work instead of value iteration to define the current policy and compute the IRL gradient. Xing et al. [4] consider path problems. To optimize the delivery path, they used the improved method heuristics in deep reinforcement learning (DRL). They analyse the aspects of delivery such as time constraints, timeliness, and high coordination. Also, they compare their method with the traditional tabu search algorithm.

Ding et al. [5] consider an alternative to traditional delivery: delivery by the crowd. Based on public transport, they develop a crowdsourcing delivery system, and they consider multiple aspects like multi-hop delivery, time constraints, and profits. From massive package data and passenger data, reinforcement learning is used to learn optimal order-dispatching strategies. Bozanta et al. [6] propose a model incorporating the order destination, order origin, and courier location. Every courier has a task to gather the given order and deliver it to the wanted place. This model is designed to increase income from served requests with a limited number of couriers over a period of time. The model of the Markov decision process is considered to simulate an actual food delivery service. The model has been applied to Q-learning and double deep Q-networks. Jahanshahi et al. [7] propose a model that employs deep reinforcement learning algorithms to solve a meal delivery service problem. The primary objective is to increase total profit by giving orders to the most suitable couriers to reduce expected delays or by postponing or rejecting orders. Their results show that the model significantly enhances the overall quality of service by incorporating the restaurants' geographical locations, customers, and the depot. Zou et al. [8] propose a reinforcement learning double deep Q-network (DQN) framework that gradually learns and tests the dispatch order policy by communicating with an online-to-offline (O2O) business simulation model designed by SUMO. Hu et al. [9] consider the dispatch problem in instant delivery services, where they dispatch a large number of orders to a small number of couriers, especially during peak hours. Based on multi-agent actor–critic, the method of courier displacement reinforcement learning (CDRL) is proposed to solve this problem. The method not only can balance supply (courier's capacity) and demand (picking up orders), it can also enhance the effectiveness of delivering orders by decreasing

idle displacing time. Table 1 presents a summary of some relevant previous studies on online food delivery service.

**Table 1.** Summary of some relevant previous studies on online food delivery service.

| Author | Method | Domain | Methodology | Objective |
|---|---|---|---|---|
| Hu et al. [9] | CDRL | Delivery services | Multi-agent actor–critic | Consider the dispatch problem of courier and balance the demand (picking up orders) and supply (couriers' capacity) and improve the efficiency of order delivery by reducing idle displacing time. |
| Zou et al. [8] | | Delivery services | Double deep Q-network | Propose a dispatch framework that gradually learns and tests the dispatch order policy by assigning the order to the selected courier, and if the order is finished within the time limit, a positive reward is given for the action. Otherwise, a negative reward is assigned for the action. |
| Jahanshahi et al. [7] | | Delivery services | Deep reinforcement learning | Consider a meal delivery service and increase the total income and quality of the service by giving the orders to the most suitable couriers in order to reduce the expected delays or by postponing or rejecting orders. |
| Bozanta et al. [6] | | Delivery services | Q-learning and double deep reinforcement learning | Consider food delivery service and increase the income derived from served orders for a limited number of couriers over a time period by utilising the order destination, order origin, and courier location. |
| Ding et al. [5] | | Delivery services | Reinforcement learning | Consider the delivery service problem and develop a crowdsourcing delivery system that uses public transport. This system incorporates some factors that impact the system, such as multi-hop delivery, profits, and time constraints. |

### 2.2. Privacy Preservation

Many scholars and experts have recently conducted much research on location and trajectory privacy preservation. Most research concentrates on the method of anonymity, and there is fairly little research on differential privacy (DP). The model of k-anonymity is commonly used by researchers to achieve trajectory protection in anonymity. The following will demonstrate the status of the research in preserving privacy by using anonymity and differential privacy (DP) [10].

Zhang et al. [11], by using a trusted anonymous server (TAS), developed a scheme of trajectory privacy preservation that aims to not allow a location-based service provider (LSP) to perform an inference attack. A group of requests generated by TAS meets the spatial k-anonymity of the user group. The TAS is the continued query that checks if the user is going to leave his/her security zone and determines if the group request needs to be resent in order to decrease the chance that the LSP rebuilds the actual trajectory of the user. Tu et al. [12] focus on semantic attacks, as if the data of the trajectory are published without appropriate handling, it could lead to severe privacy leakage, and existing solutions do not provide adequate protection to protect against semantic attacks. This means the attacker could obtain an individual's private information by using the semantic features of frequently visited locations on the trajectory. Therefore, they propose an algorithm that provides high privacy safeguards against semantic attacks and re-identification while keeping the data utility at the highest level. Chiba et al. [13] propose an algorithm that over a certain range, when the position information is acquired, the algorithm mismatches the time with the position information. They defined indicators that represent position distortions and information about the time. Zhou and Wang [14] propose a defence algorithm based on k-anonymity and fog computing. A scheme of trajectory protection is designed for the protection of offline data in trajectory publication and the protection of real-time trajectory data with continuous queries. Their solution incorporates the mobility and local storage provided by fog computing to guarantee physical control along with a cloaking region constructed by k-anonymity for each snapshot.

The DP model has been favoured by many scholars as it has a rigorous mathematical background. With DP and by adding noise to DP data, attackers cannot determine if the database contains data records, which achieves privacy protection purposes. Moreover, in machine learning, privacy attacks are usually addressed by using the technique of differential privacy [15]. Andrés et al. [16] propose a formal privacy notion of geoind for location-based systems that protect the user's exact location while permitting the release of the approximate information that is usually required to get a certain service. By adding to the user's location managed random noise, geoind achieves privacy preservation. Deldar

and Abadi [17] study how to not increase the risk of a privacy breach and how different geographical map locations can meet the requirements of individual user privacy protection. To achieve user location privacy preservation, the personalized location differential privacy (PLDP) concept was introduced for trajectory databases. A personalized noisy trajectory tree is used by PLDP-TD and is built from the underlying trajectory database to provide a response to statistical queries by using the differential privacy method. Zhao et al. [10] proposed, based on clustering and using DP, a novel trajectory-privacy-preservation method. In the cluster, to prevent an attack of continuous queries, Laplace noise is added to the count of the trajectory location. The radius-constrained Laplace noise is added to the trajectory location data in the cluster to avoid too much noise affecting the clustering. The noise clustering centre in the cluster is obtained according to the noise location data and the count of the noise location. Yang et al. [18] consider the issue of user location privacy protection. As the centralized server is required to obtain each user's location precisely to ensure optimal task allocation, this raises a privacy concern of exposing the workers' exact locations. To tackle this issue, a crowdsensing data release mechanism that meets DP is proposed to provide strong protection of worker locations. Table 2 presents a summary of some relevant previous studies on privacy preservation.

**Table 2.** Summary of some relevant previous studies on privacy preservation.

| Author | Method | Domain | Methodology | Objective |
|---|---|---|---|---|
| Andrés et al. [16] | geoind | Location privacy | Differential private | Consider the location privacy problem and protect actual location information. Propose a formal privacy notion of geoind for location-based systems that protects the user's exact location and permits the release of approximate information that is usually required for certain services. |
| Deldar and Abadi [17] | PLDP | Location privacy | Differential private | Consider location privacy and guarantee that the privacy of each sensitive location or moving object is protected. |
| Zhao et al. [10] | | Location privacy | Differential private | Consider protecting the location and trajectory information privacy from being disclosed by attackers. Based on clustering using differential privacy, proposed a trajectory-privacy-preservation method in order to protect trajectory information. |
| Yang et al. [18] | | Location privacy | Differential private | Consider location privacy and the privacy of the workers' location information. Proposed data release mechanism crowdsensing techniques that aim to protect the privacy of the worker location information. |

### 2.3. Discussion of Related Work

Much research tackles the food delivery problem by using different methods, such as assigning the delivery request to the most appropriate courier, trying to minimise the number of rejected requests, or proposing an effective dispatch mechanism. In this work, we consider this issue and employ multi-agent reinforcement learning. We used two novel algorithms—QMIX and IQL—to guide the courier to the area with high order demand to increase the number of orders received by the courier and raise his/her long-term income.

While improving the efficiency of food delivery services is necessary, preserving the privacy of the users' locations is crucial. Numerous researchers have studied this issue, and methods such as k-anonymity and DP have been used to protect the user's location. Our work proposes the PULM method to protect the user's location. The PULM employs the DP mechanism and injects Laplace noise into the user location along with the courier trajectory. This method considers two critical factors—the city area size and the frequency of customers ordering online food delivery—in order to identify the amount of injected noise.

## 3. Preliminary

### 3.1. Dueling Network Architecture

Dueling DQN is a reinforcement learning model-free network that solves the Bellman Equation iteratively [19]. It was proposed in 2015 by Wang et al. [20] as an improvement to DQN. The main insight of this new architecture is that there is no need to estimate the value of each action choice for many states. For example, knowing whether to move left or right in the Enduro game setting only matters when a crash is close. It is essential to know which action to take in some states, while in other states, the chosen action does not

affect what happens [20]. In this algorithm, the main improvement is that the Q-values $Q(s, a)$, which the network tries to approximate, can be split into two parts: the advantage of actions of the state $A(s, a)$ and the value of the state $V(s)$. Based on the definition $Q(s, a) = V(s) + A(s, a)$, the advantage $A(s, a)$ is assumed to bridge the gap from $A(s)$ to $Q(s, a)$. We can suppose the advantage $A(s, a)$ is delta, which tells us how much reward we can earn from each specific action in each particular state. In general, the advantage can be negative or positive and can have any magnitude [21]. There is no sequential architecture such as deep learning in dueling DQN. The model layers are divided into two different streams after the convolutional layers. Each stream has its own fully connected layer and output layers [22]. The outputs of the two separate estimators can be integrated as follows:

$$Q(s, a; \theta, \alpha, \beta) = V(s; \theta, \beta) + \left( A(s, a; \theta, \alpha) - \frac{1}{|\mathcal{A}|} \sum_{a'} A(s, a'; \theta, \alpha) \right)$$

The parameters of the convolutional layers are denoted by $\theta$, while $\alpha$ and $\beta$ are the parameters of the two streams of fully connected layers.

### 3.2. Multi-Agent Reinforcement Learning

One of the widespread solutions for sequential decision-making problems is reinforcement learning. In the setting of multi-agent reinforcement learning (MARL), we move to the problem of having more than one agent in the environment and abandon the problem of having one single agent. MARL is about the intersection of two concepts: the task is achieved by reinforcement learning, and agents interact with other agents [23]. In this setting, we have more than one agent interacting in our environment and aiming to maximize our rewards [24].

The Markov decision process is used in fully observable domains as a model of MARL and can be described as six tuples containing: (1) $I = \{1, \ldots, j\}$, representing the set of agents, where $j$ is the number of the agent; (2) the environment true state $S$ at each time step; (3) $A = \{a_1, \ldots, a_n\}$, the agent's available actions, where $n$ is the action index; (4) the transition function $T$, which depend on the agent's action; (5) the agent reward function $R$; and (6) $\gamma$, the discount factor of every agent. The domain at each step $t$ is in state $s \in S$; The agent takes action $a \in A$; this action transits the domain to a new state $s'$ with probability $T(s'|s, a)$; the agent then gets a reward $r$ based on $R(s, a)$; this process is repeated until the agent stops. The agent's main objective is to learn a policy $\pi$ that will raise its expected discounted future reward $E[\sum_{0 \leq t < \infty} \gamma^t r^t]$ as the number of sequential steps is $t$ [23,25,26].

#### 3.2.1. QMIX Algorithm

In a multi-agent framework, every agent selects an action, creating a collective action $a_t$, and then the global, immediate reward $r_t$ is shared, which assesses the collective action taken previously. For the collective action, there is collective agent–value function $Q_{tot}(s_t, a_t) = \mathbb{E}_{s_{t+1}: \infty, a_{t+1}: \infty}[R_t|s_t, a_t]$, where at time $t$, the discounted return is $R_t$. One of the main challenges in MARL is how to assess every agent's contribution individually and precisely to get from the collective action–value function to the individual value function. The agent $A_i$ individual value function is represented by $Q_i(o_t, a_t)$. QMIX is an advanced technique of value-based aims to train uncentralized policies in an end-to-end centralized method [27]. To perform this, QMIX needs to overcome two challenges: The first challenge is in the process of centralized training. Every agent needs to calculate the influence action based on a single global reward. The second challenge is to guarantee that the ensemble of the optimal actions of agents is the optimal action of the ensemble of agents when agents interact in a decentralized manner [28]. There is a belief in QMIX that it can interpret the total action–value function as the mixing network of each action–value function or as a linear combination of each action–value function. Thus, it is expected that the QMIX-based method acts in cooperation with nearby agents [29].

### 3.2.2. Independent Q-Learning

Independent Q-learning (IQL) was proposed by Tampuu et al. (2017) [30]. Independent Q-learning is an algorithm that abandons centralized training: for each agent in IQL, Q-learning is performed separately. In some way, this method avoids the implementation issue of the CTDE framework. Consequently, there is a high probability that a single agent action can interfere with the overall environment and with other agents of the state. Therefore, this makes it so that IQL cannot converge in a complex environment. The IQL algorithm still has a number of applications in small scenarios of reinforcement learning applications such as Atari games.

In IQL, each agent trains decentralized Q-functions, while QMIX is a method learned from end-to-end for decentralised policies in a centralised setting. QMIX is composed of networks of agents representing every $Q_a$ and a mixing network that combines them into $Q_{tot}$. IQL is the most commonly applied method in multi-agent learning, and it consists of a group of concurrent single agents that use and share the same environment but where every agent learns individually [27,29,31,32]. In our research, we intend to use different MARL settings in order to show more results from our experiments.

### 3.3. Trajectory Protection

Assume the universe of locations is $\mathcal{L} = \{L_1, L_2, \cdots, L_{|\mathcal{L}|}\}$, $|\mathcal{L}|$ is the size of the universe, the locations are discrete spatial areas on a map, and trajectories are formed from series of order locations that are drawn from the universe.

**Definition 1** (Trajectory). *The trajectory T of length $|T|$ is an ordered list of locations $T = t_1 \rightarrow t_2 \rightarrow \cdots \rightarrow t_{|T|}$, where $\forall_1 \leqslant i \leqslant |T|$.*

In $T$, the locations can occur consecutively and can occur multiple times. Therefore, a correct trajectory can be $\mathcal{L} = \{L_1, L_2, L_3, L_4\}$, T = $L_1 \rightarrow L_2 \rightarrow L_3$. The timestamps, in some cases, can be included in the trajectory. The trajectory database consists of multiple trajectories, and the record owner of the movement history is shown for every trajectory. The following is the formal definition:

**Definition 2** (Trajectory Database). *A trajectory database D of size $|D|$ is a multiset of trajectories $D = \{D_1, D_2, \cdots, D_{|D|}\}$.*

Table 3 shows a sample of database trajectories with $\mathcal{L} = \{L_1, L_2, L_3, L_4\}$ [33]. The attacker may get the user's location via analysis of trajectory data. In this research, a defence method based on DP is provided for some attack models existing in trajectory analysis. A properly calibrated randomization mechanism is injected with a meaningful amount of differential privacy noise drawn from a trajectory sensitivity function as described in Section 3.4. This creates differential privacy spatio–temporal data; the sanitization of a trajectory path is the result of the perturbation of these traces. Therefore, the perturbed trajectory path prevents the attacker from determining the original trajectory path and preserves user privacy.

**Table 3.** Sample trajectory database

| Rec # | Path |
| --- | --- |
| 1 | L4 → L5 → L6 |
| 2 | L2 → L3 |
| 3 | L2 → L4 → L5 |
| 4 | L3 → L5 → L6 |
| 5 | L4 → L5 |
| 6 | L1 → L3 |
| 7 | L3 → L2 |
| 8 | L1 → L3 → L5 → L2 |

*3.4. Differential Privacy*

Today, with the advancement of data analysis and mining, data privacy threats are vastly increasing. Differential privacy (DP) is mainly a mathematical model that ensures the privacy of a statistical dataset [34]. Dwork et al. [35,36] provide robust standards to maintain privacy in data analysis. With no previous knowledge of the trader's background knowledge, DP is the standard model for privacy preservation that is used to maintain privacy [37]. Let $\mathcal{X}$ be considered a finite data universe. The variable $d$ is attributes of universe $\mathcal{X}$, and $r$ is the sample record. From domain $\mathcal{X}$, $D$ is a set of $n$ records that are unordered. If $D$ and $D'$ datasets are different in one record only, then they are considered neighbouring datasets. The function $f$ is a map of a query of the $D$ dataset to an abstract range $\mathbb{R} : f : D \rightarrow \mathbb{R}$. To maintain privacy, we mask the difference of query $f$ between the neighbouring datasets. In DP, the sensitivity $\Delta f$ is an essential concept that represents the result of the highest difference of query $f$, which defines how much perturbation is needed for the private-preserving answer. The algorithm of randomization $M$ is used to perform this goal as it reaches the database and executes several functions. The following is the definition of DP [38]:

**Definition 3 (Differential privacy).** *A randomized mechanism $\mathcal{M}$ providing differential privacy $(\epsilon, \delta)$ for any two neighbouring datasets $D$ and $D'$ for which for every set of output $\mathcal{Z}$, $\mathcal{M}$ satisfies:*

$$\Pr(\mathcal{M}(D) \in \mathcal{Z}) \leq \exp(\epsilon) \cdot \Pr(\mathcal{M}(D') \in \mathcal{Z}) + \delta \tag{1}$$

The randomized mechanism $\mathcal{M}$ provides its strictest definition if $\delta = 0$.

**Definition 4 (Sensitivity).** *for neighbouring inputs $(D , D' \in \mathcal{D})$ and query $f : D \rightarrow R$, the sensitivity $f$ is defined as:*

$$\Delta f = \max_{D,D'} ||f(D) - f(D')|| \tag{2}$$

Sensitivity is the parameter that defines how much perturbation is needed for a certain query. The highest difference between the results of the query on neighbouring datasets is considered by the sensitivity $\Delta f$.

**Definition 5 (Laplace mechanism).** *for the function $f : D \rightarrow R$ with dataset $D$, the mechanism $\mathcal{M}$ gives $\epsilon$ differential privacy [37,39,40].*

$$\mathcal{M}(D) = f(D) + Lap(\frac{\Delta f}{\epsilon}) \tag{3}$$

The mechanism of Laplace is used with numeric outputs, as the mechanism inserts independent noise into the original answer.

**4. Methods**

*4.1. Model Overview and Problem Statement*

In most cities, particularly large cities, the demand volume varies from one particular area to another and from time to time. To increase the number of received online food delivery orders, many food delivery couriers try to locate areas with high food delivery demand based on their experience. However, sometimes they may go to an area with low food delivery demand and receive fewer food delivery requests.

The main objective of our proposed model is to increase the number of received orders by the courier and thereby increase the courier's long-term income. Our method can minimize the waiting time in rush hours, as there will be enough couriers in areas with high demand for food orders. Figure 1 shows the flowchart showing the steps of execution of the online food delivery service method. We consider urban areas and specific areas of cities, which we then divide into multiple parts. We assume this area to be a rectangular

shape that can be split into $N \times M$ cells. Every divided cell illustrates a small area in the city, as demonstrated in Figure 2. The yellow cars represents the couriers, and the red circles represent areas with high demand for food delivery requests. A courier of food delivery in a cell waits and receives a delivery request from a customer without any previous data, as customer data are available only after the customer requests the service. The courier has to learn to choose a cell with high food delivery requests based on analysis of the data collected by an agent. Each time, the courier tries to improve by selecting the most appropriate cell to increase the number of received orders and increase long-term income.

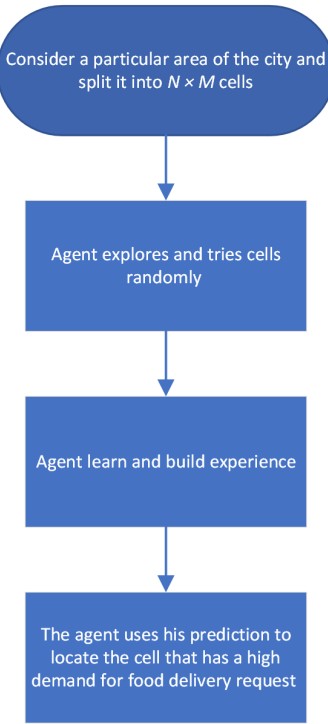

**Figure 1.** Flowchart showing the steps of execution of online food delivery service method.

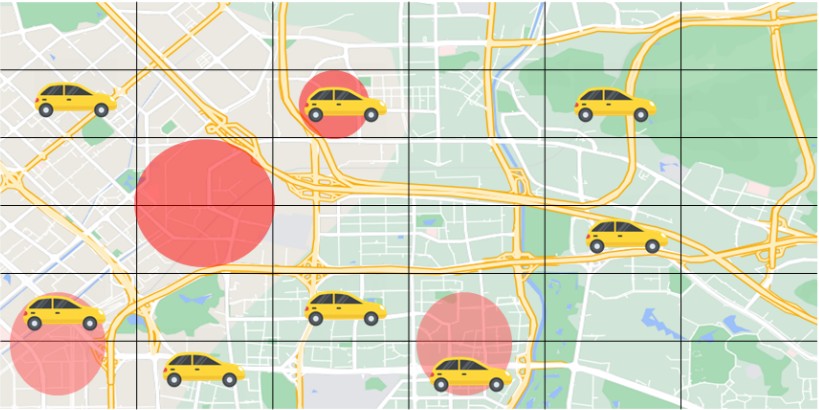

**Figure 2.** Overview of the model: the yellow cars represent the delivery cars that are being guided, and the red circles represent areas with high demand for food delivery requests.

While enhancing online food delivery services is important, protecting customer information is essential. This research considers preserving customer privacy information by protecting customer location information. We utilize the DP Laplace mechanism, where the privacy parameter $\epsilon$ that affects the amount of injected noise depends on two parameters: the city area size and the frequency of customers' online food delivery orders. To protect

the customer location information, we inject Laplace noise into the user locations and courier trajectories.

### 4.2. Multi-Agent Reinforcement Learning Formulation

The Markov decision process is used to formulate our model. The transition probability in our case is unknown. The idea of being model-free is more appropriate for such situations. Below are more details about the main components of the model and the Markov decision process formulation.

#### 4.2.1. Agent

In our environment, more than one agent has been created $\{A_1, A_2, \ldots, A_n\}$, so we must allocate the learning policy to each agent respectively. This MARL problem considers the courier as the agent. The agent's main goal is to pick a cell with high demand of food delivery requests, where the courier can increase the number of received food delivery requests and increase the courier's income. Initially, the agent policy is uncertain and requires interaction with the environment to learn. The agents evaluate the policy and iterate consecutively.

#### 4.2.2. State

The global state $s_t$ is maintained at each time $t$, and the couriers' spatial distribution is considered. In our dynamic model, the states are defined based on grid location and time. The state space for our problem is characterized as follows:

- The grid location: A certain area of the city map is considered and is divided into $N \times M$ cells. Every divided cell illustrates a small area in the city. The agent interacts with the environment and updates his/her status. The agent then analyses and computes the available information about the environment and tries to pick a cell with high food delivery demand.
- Weekdays and weekends: On the one hand, during weekdays and weekends, different situations need to be noted and considered by the model. Therefore, in this model, we consider weekdays and weekends during the learning process for better performance.

#### 4.2.3. Action

For each state, the agent chooses the cell that makes the courier get more food order requests and increase long-term revenue. The map is split into $N \times M$ cells, and each cell represents a particular area that may or may not have restaurants. The model calculates the number of orders in each cell and analyses the data based on that.

The agent has to perform an action on this by picking the most proper cell to raise his/her rewards. In this case, the agent has $N \times M$ actions and needs to choose one cell each time.

#### 4.2.4. Reward Function

The agent in the environment performs the action, and then the agent receives the rewards from the environment. Only if the agent is able to increase the number of received orders and meet or exceed the approximate average number $A$ is the received reward positive; otherwise, the reward is negative. This can be defined as follows:

$$r(k) = \begin{cases} 1, & if \; \mathcal{N} \geq A \; . \\ 1-, & \text{Otherwise} \; . \end{cases} \tag{4}$$

where $r(k)$ is the reward function, $\mathcal{N}$ is the number of orders, and $A$ is the approximate average number of orders for every single agent.

### 4.2.5. State Transition

In our setting, the action is passed from agents and is received by the environment. The state $s_t$ is then generated, the agents observe the new state, and the agents take collective action $a_t$. The environment assesses this action, and the environment returns a reward $r_t$. A tuple of state transition $\{s_t, a_t, r_t, s_{t+1}\}$ is formed at this point, and then MARL is used to find the relationships between these tuples.

### 4.2.6. Multi-Agent Reinforcement Learning for Online Food Delivery

An overview of the QMIX algorithm for online food delivery is presented in Algorithm 1. During the training process and by using gradient descent, the loss function is minimized with consideration of the parameter $\theta$ at iteration $i$. For this purpose, a neural network is employed as an approximator. In the beginning, with size $\mathcal{N}$, we initialize the reply buffer $\mathcal{D}$. In lines 2 to 4, with random weight $\theta$, we initialize the action value function, copy weight $\bar{\theta} \leftarrow \theta$, initialize the target action value function, and then initialize the QMIX mixing network. In lines 5 and 6, if the episode has not ended, the environment then resets. In lines 9 to 11, the current observation is observed by agents, and the agent then chooses an action based on probability $\epsilon$ or else chooses the highest value from $argmax_{a_t}(o_t, a_t : \theta)$. In line 13, the agent performs the action and obtains corresponding rewards; then, the agent observes the next state. In line 14, experience tuples, such as the taken action, state, next state, and obtained reward, are inserted to memory. From the reply buffer $\mathcal{D}$, we extract the sample and compute the loss function $\mathcal{L}$. In line 18, we implement the step of gradient descent. In lines 19 and 20, the parameter $\bar{\theta}$ of the target network and the exploration rate $\varepsilon$ are updated.

---

**Algorithm 1** QMIX algorithm for online food delivery environment.

---

1: **With** size $\mathcal{N}$, initialize reply buffer $\mathcal{D}$
2: **With** random weight $\theta$, initialize action value function
3: **With** $\bar{\theta} \leftarrow \theta$, initialize target action value function
4: **Mixing** network $\phi$ is initialized
5: **while** Episode training not finished **do**
6:     Environment reset
7:     **while** the state not in termination **do**
8:         **for** agent in Agents **do**
9:             Observing the state $o_i^t$ by the agent
10:             Select a random action $a_t$ with probability $\varepsilon$
11:             Else select $argmax_{a_t}(o_t, a_t : \theta)$
12:         **end for**
13:         Perform action $a_t$, move next state $s_{t+1}$, and obtain reward $r_t$ and termination information
14:         Insert $(o_t, a_t, s_{t+1}, r_t)$ to $\mathcal{D}$ reply buffer
15:         Random sample $\mathcal{K}$ minibatch from $\mathcal{D}$
16:         $Q_{tot} = mixing((Q_1(o_t^1, a_t^1) + Q_2(o_t^2, a_t^2) + ...Q_n(o_t^n, a_t^n))$
17:         Loss function computed $\mathcal{L} = \left(r_t + \gamma max_{a_t} Q_{tot}(s_t, a_t | \bar{\theta}) - Q_{tot}(s_t, a_t | \theta)\right)^2$
18:         On $\mathcal{L}$, implement the step of gradient descent into network parameter $\theta$
19:         Updating the $\bar{\theta}$ parameter of the target network
20:         Rate $\epsilon$ of exploration is updated
21:     **end while**
22: **end while**

---

### 4.3. Protect User Location Method

Safeguarding the privacy of customer locations in online food delivery applications is essential. In this section, we focus on how to safeguard the user location in a delivery application. Therefore, we designed the PULM method to protect the user's location by injecting noise into the user's location and the courier's trajectory. Figure 3 is a flowchart showing the steps of execution of the PULM method. We use DP along with the Laplace mechanism to guarantee that the user's location is not disclosed. We determined the privacy

parameter in this method based on two parameters. The first parameter is the city area size and the second parameter is the customers' frequency of online food delivery orders.

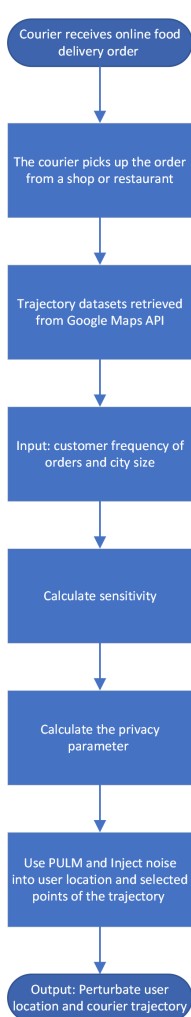

**Figure 3.** Flowchart showing the steps of execution of the PULM method.

### 4.3.1. Overview of PULM Algorithm

Preserving the privacy of user locations is crucial and essential. DP shows robustness and effectiveness in providing adequate privacy to datasets. Algorithm 2 shows the execution of the PULM method and how it works. In the beginning, the trajectory datasets of the courier are entered in the form of location points containing longitude and latitude from the start point to the destination. In lines 2 and 3, the frequency of a customer's ordering and the city area size are calculated and entered. The sensitivity $\Delta f$ is calculated in line 4 and is based on Equation (2). In line 5, the privacy parameter $\epsilon$ is calculated. To calculate the privacy parameter, we sum the result of the city area size with a weight of 2.5 with the result of the frequency of online food ordering with a weight of 2.5; a smaller value for the privacy parameter is stronger privacy. For example, the Iowa City area $<500$ km$^2$ gives 0.5. If we have a customer with five orders and in the dataset $\max(x) = 10$, $\min = 1$, this will be calculated as $(5–10)/(1–10) \times 2.5 = 1.388$. In this case, the privacy parameter is $0.5 + 1.388 = 1.88$. In line 6, random points are taken from the trajectory dataset, including the location of the customer. In line 7, Laplace noise is added based on the sensitivity $\Delta f$ and privacy parameter $\epsilon$. The output perturbates the user location with the trajectory.

---

**Algorithm 2** Protect User Location Method (PULM)

---

1: **Input** user trajectory datasets $\mathcal{D}$
2: **Input** customer frequency of ordering online food delivery.
3: **Input** city size
4: Based on Equation (2), obtain the sensitivity $\Delta f$ of the trajectory metric space
5: Calculate the privacy parameter $\epsilon$ based on city size and customer order frequency
6: Select random points from the input trajectory dataset $\mathcal{D}$, including the user location
7: Add Laplace noise to the selected points based on $\Delta f$ and $\epsilon$
8: **Output** Perturbate user location with trajectory

---

### 4.3.2. City Area Size

The city area size is an important factor, as it can affect the ability to infer the user location and courier trajectory. Figure 4 shows the difference between the two examples of city area size that we consider in this research. For a large city area size, the population, the number of roads, and the number of cars are more. Therefore, inferring the trajectory and customer location may be more difficult compared with a smaller city area size. For example, Figure 4 shows the sizes of Iowa City and Shenzhen, where the area of Iowa City is about 67.83 km$^2$ compared with about 2050 km$^2$ for Shenzhen; the population in Iowa City is about 74,596 compared with about 17.56 million in Shenzhen.

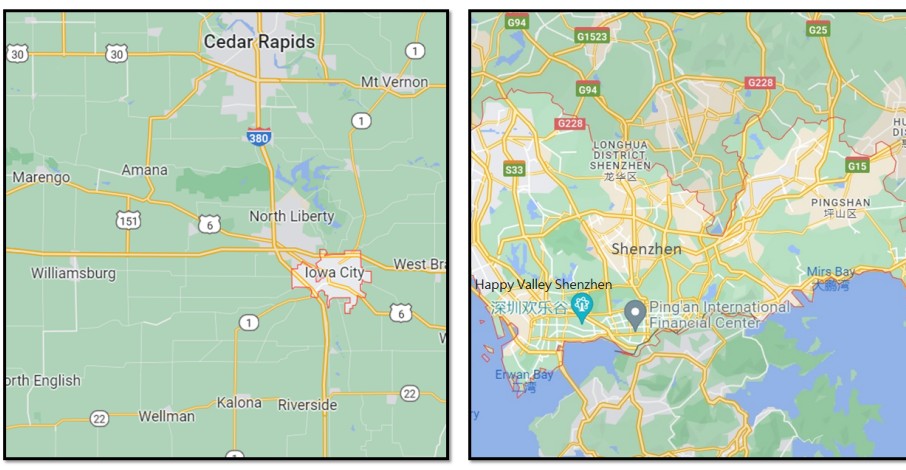

**Figure 4.** Comparison of the city size of Iowa City on the left and Shenzhen on the right side.

The city area size varies from 50 km$^2$ to more than 8000 km$^2$, such as Tokyo with 2191 km$^2$ city area size [41] and a small city such as Iowa City with 66.3 km$^2$ [42]. To allocate the proper noise in our model, we need to classify the city based on the area size. Most of the city classifications are based on the city population. For example, UK Parliament [43] classified cities based on population into core cities such as London, Sheffield, and Glasgow; other cities; large towns; medium towns; small towns; and villages. Also, Liu and Chen [44] classify cities as super cities for populations of more than 10 million, megacities for populations between 5 and 10 million, large cities for populations between 1 and 5 million, medium cities for populations between 0.5 and 1 million, and small cities for populations less than 0.5 million. In our model, we attempted to classify cities based on area size, and we approximately mimicked the population factor. Equation (5) shows the distribution of noise parameters based on the city area size that we used in our model.

$$f(x) = \begin{cases} 0.5, & \text{if } x <= 500. \\ 1, & \text{if } 500 < x <= 1000. \\ 1.5, & \text{if } 1000 < x <= 1500. \\ 2, & \text{if } 1500 < x <= 2000. \\ 2.5, & \text{if } x > 2000. \end{cases} \tag{5}$$

### 4.3.3. Customer Frequency of Online Food Delivery Orders

In our method, we consider the factor of how frequently a customer makes orders. When there are many orders for the same customer, there is a high chance of inferring the information of the customer. The attacker may be able to infer the customer's location and pattern or any other information, especially if there are many repeat orders from the same user. An example of the ability of the attacker to infer some private information from a particular user is that the attacker could infer if a particular individual has health issues if the user frequently visits a hospital location that is not his home or workplace [12]. Consequently, we consider this factor and aim to protect the user location by injecting more noise into the user locations and courier trajectories for users with high-frequency orders. We calculate the privacy parameter based on two parameters: the city area size and the frequency of a customer's order. The frequency of orders for the customer can be determined using this equation:

$$\mathcal{Y} = ((x - Max(x)) \,/\, (Min(x) - Max(x))) \times 2.5 \tag{6}$$

where $x$ represents each customer's frequency of order number, $Max(x)$ is the dataset's maximum frequent order number, and $Min(x)$ is the dataset's minimum frequent order number.

## 5. Experiment Design

In this section, we show the experiment setup for multi-agent reinforcement learning for online food delivery services built based on multi-agent reinforcement learning. Furthermore, we show the setup for preserving the privacy of customer locations and the details about the datasets we used in these experiments.

### 5.1. Multi-Agent Reinforcement Learning for Online Food Delivery Services

In this part, we present breakdown details on how each experiment of online food delivery methods is set up. We deliver some experiments to show how the agent can accumulate rewards by using different algorithms of reinforcement learning on various datasets and how much the agent can increase the number of food delivery orders over time. We considered the weekdays and weekends during the learning process for better performance, and we assume the first record in the dataset is the first day of the week.

### 5.1.1. Applying Different Algorithms for Deep Reinforcement Learning

An extensive experiment was done to show the results of using multi-agent reinforcement learning in online food delivery methods. Two MARL algorithms were used along with one single agent. We used QMIX and IQL to implement MARL and dueling deep Q-networks to implement the single agent algorithm. We tried to simulate the results from a single agent by multiplying the results from a single agent to be equal to the number of agents in MARL. We ran these algorithms with different datasets from different cities that have different city area sizes. In each episode, the agents were run multiple times. The used datasets were Shenzhen, China; and Iowa City, USA. Moreover, we applied the algorithm to a synthetic dataset (random data) to show more comparisons and the effects of our proposed method.

### 5.1.2. Number of Food Delivery Orders

In this experiment, we implement multi-agent reinforcement learning for the online food delivery method, showing the number of orders the courier can gain and how much the courier's orders can increase over time. QMIX was used on two datasets: Shenzhen and Iowa City. The results of the total number of orders for three agents were shown. The experiment demonstrates to what extent all agents can gain more orders in Shenzhen and Iowa City datasets.

### 5.2. Trajectory Protection Method

Preserving location privacy for the user is vital. In this experiment, we used the PULM, which employs a DP Laplace mechanism to preserve the user location. We inject noise into the customer location along with the courier trajectory. The method considers two important factors to determine the privacy parameter $\epsilon$. The first parameter is city area size and the second parameter is customer frequency of online food orders. These factors determine the amount of injected noise based on the DP algorithm. To evaluate our approach, we use Shenzhen, China; and Iowa City, Iowa, USA; datasets. We use the Google Maps API to retrieve the trajectory by sending the start and end points and then retrieve the trajectory of the courier. We then use the PULM method to obfuscate the trajectory by injecting noise into selected points of the trajectory.

### 5.2.1. Trajectory Data Utility

Adding Laplace noise to the trajectory inevitably affects the data utility. In order to evaluate the data utility in the trajectory after adding noise, we used the Hausdorff distance (HD). The Hausdorff distance is a widely used metric to measure the similarity of two datasets of points and allows the measurement of the utility of dataset $D$ with respect to dataset $\tilde{D}$ [45]:

$$utility(D) = max(h(D, \tilde{D}), h(\tilde{D}, D)) \tag{7}$$

This experiment demonstrates the results of the Hausdorff distance (HD) for Shenzhen, China; and Iowa, USA; datasets, along with the average of the Hausdorff distances for both datasets.

### 5.2.2. Analysing Privacy Parameter Intensity

In this part, the distribution of privacy parameters generated by PULM is analysed. The privacy parameter is changed frequently based on the city area size and the customer frequency of online food orders for each customer. The privacy parameter affects the amount of noise generated by the Laplace mechanism. We analyse the generated privacy parameter based on two datasets: Shenzhen, China; and Iowa City, Iowa, USA.

### 5.3. Dataset Description

Two different datasets with different city area sizes were used for our experiments to test the approach that applies to the online food delivery service and the Protect User Location Method (PULM).

The first dataset is provided by Alibaba Local Service Lab [5] and contains on-demand food delivery order data. The dataset for Shenzhen, China, has 1,048,575 orders and 93 restaurants along with the location of the restaurants, delivery locations, the times of pickups of meals from restaurants, and the delivery times. The second dataset used in our experiment is provided by Ulmer et al. [46]. This dataset contains data of meal delivery services in Iowa City, Iowa , USA. In this dataset, there are 111 restaurants and 1,200,391 delivery records. In this dataset, the actual locations are used with random generation of orders and equal request probability for each point of time and location. For both cities, we consider a particular area to implement our experiment. The data may have been modified from the source for certain purposes, such as obfuscating exact locations.

## 6. Results

*6.1. Multi-Agent Reinforcement Learning for Online Food Delivery Service*

In multiple domains, DRL has been proven to resolve complicated sequential decision-making problems [28]. In this, we show our results by using multi-agent reinforcement learning techniques QMIX and IQL along with a single agent of dueling deep Q-networks on different varied datasets.

Figure 5 shows the results of the average accumulated reward by using QMIX in black, IQL in blue, and a single agent of dueling deep Q-networks in red. The Shenzhen, China, dataset was used in this experiment, and we show the results of different runs of 5000, 10,000, and 20,000. The agents in this experiment increase cumulative rewards over time and are able to reach the highest level. The single agent increases its accumulative rewards over time and shows that it has higher results. This experiment was repeated on Iowa, USA, and synthetic random datasets as shown in Figures 6 and 7, respectively, which show that the agents can increase cumulative rewards over time and are able to reach the highest level. The double agents increase their accumulative rewards over time, and they demonstrates that they achieve higher outcomes in comparison with the single agent.

Figure 8 illustrates the average number of food delivery orders that the courier can gain after implementing multi-agent reinforcement learning for the online food delivery method using QMIX with the two datasets: Shenzhen and Iowa City. It shows that the agents were able to increase the number of orders from 80 orders to 140 orders for all agents in the Shenzhen dataset and from 10 to 40 in the Iowa City dataset for all agents.

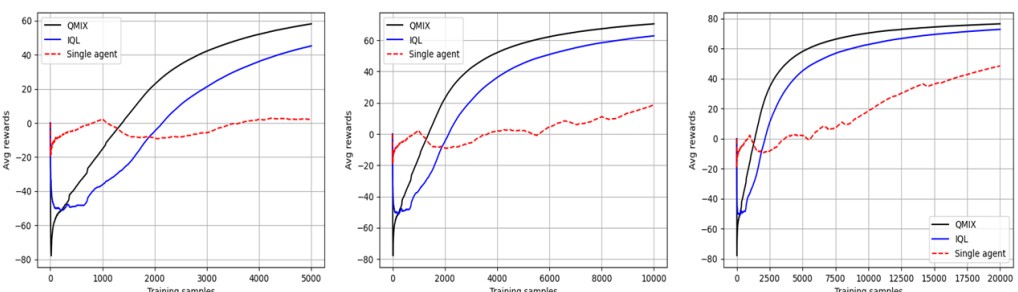

**Figure 5.** Comparison of the results of the online food delivery method using the Shenzhen dataset with two MARLs and a single agent with different numbers of runs.

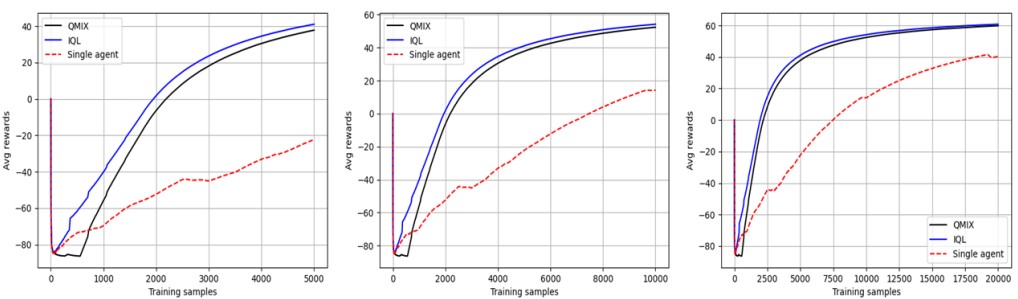

**Figure 6.** Comparison of the results of the online food delivery method using the Iowa City dataset with two MARLs and a single agent with different numbers of runs.

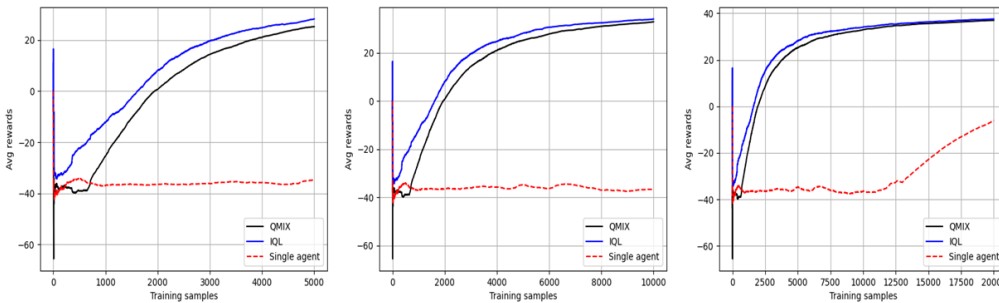

**Figure 7.** Comparison of the results of the online food delivery method using a random synthetic dataset with two MARLs and a single agent with different numbers of runs.

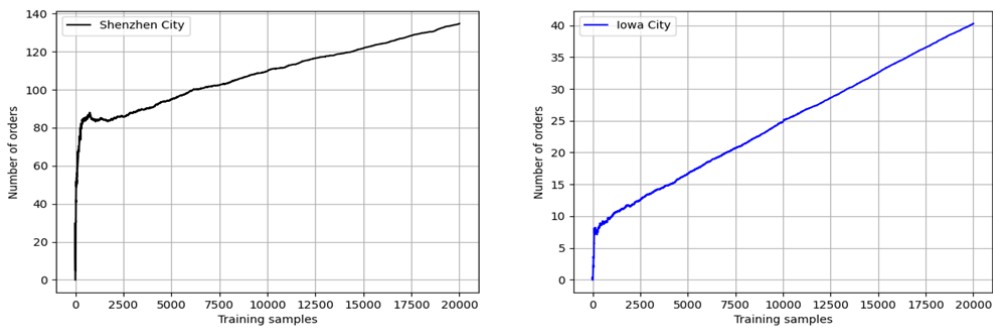

**Figure 8.** The average number of food delivery orders obtained by drivers after using MARL QMIX with the Shenzhen and Iowa City datasets.

### 6.2. Trajectory Protection Mechanism

Preserving the privacy of customer locations is vital. This experiment shows the results of using the PULM method, where we aim to protect the customer location along with the courier trajectory. We use the Google Maps API to get the trajectory by sending the start and end destination points to the Google Maps server and retrieving the trajectory. By using our approach, proper noise is injected into a selected point in order to obfuscate the trajectory and user location.

After Laplace noise is injected into the trajectory, the utility of the data is definitely impacted. To evaluate the data utility in the trajectory after adding noise and to see how much the obfuscated trajectory was impacted, we used the Hausdorff distance, which is a commonly used method to determine how much similarity exists between two datasets' points. Figure 9 shows the result of the Hausdorff Distance along with the average Hausdorff Distance in Shenzhen and Iowa. The figure shows slight similarity of HDs between the two datasets with some fluctuation that normally happens with obfuscation functions.

The PULM determines the privacy parameter $\epsilon$. In this matter, the privacy parameter changes frequently based on the city area size and the customer frequency of ordering food online. Figure 10 shows a few samples of the distribution of privacy parameter $\epsilon$ generated by the PULM algorithm over the two datasets: Shenzhen, China; and Iowa City, USA. The generated parameter can range between 0.5 to 5. The figure shows that the distribution of the privacy parameter $\epsilon$ in the Shenzhen dataset ranges from 4.25 to 4.5, and in the Iowa City dataset, it ranges from 1.3 to 3.

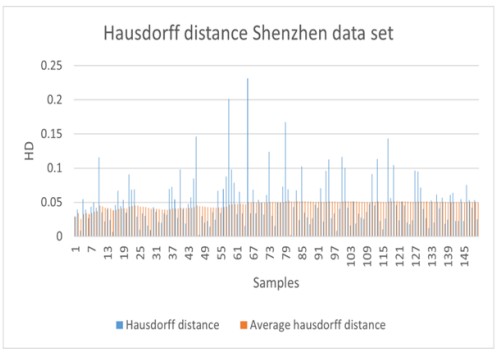
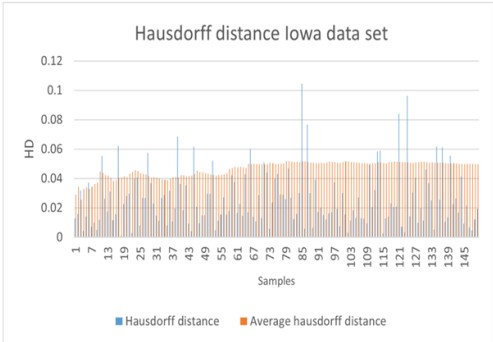

**Figure 9.** The results of using Hausdorff distance to evaluate the data utility after using PULM and adding Laplace noise to the trajectory and user location for Shenzhen and Iowa City datasets. The X-axis demonstrates the number of used samples, and the Y-axis indicates the corresponding results of the Hausdorff distance in blue and the average Hausdorff Distance in orange.

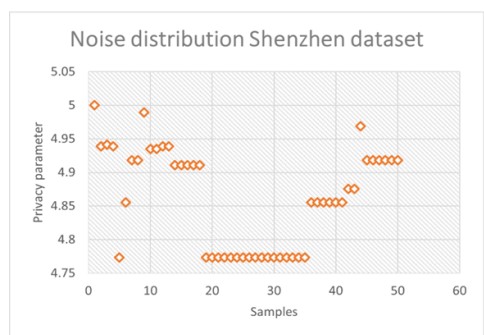
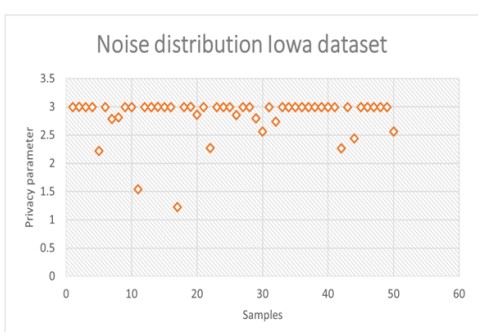

**Figure 10.** The results of the privacy parameter $\epsilon$ distribution generated by the PULM algorithm over the Shenzhen and Iowa City datasets. The *Y*-axis indicates the privacy parameter $\epsilon$, and the *X*-axis represents the samples.

## 7. Conclusions

This research considers how to increase the number of received online food orders by couriers and increase their income. Multi-agent reinforcement learning is used to train the model and guide the agents to areas with high food delivery order demand. The city map is divided into small grids, and each grid represents a small area of the city. The agent has to learn which grid has the highest demand. To protect the customer location, we propose PULM. The PULM injects DP Laplace noise based on the city area size and customer frequency of ordering food online. We used two datasets—Shenzhen, China; and Iowa, USA—to demonstrate our experiment results.

## 8. Future Work

Although this research proposes a solution to online food delivery services and the related privacy issues, there is still space to improve this work and develop different methods that can achieve better results. There is an excellent opportunity to propose a new multi-agent reinforcement learning method or to improve the current work to enhance the learning rate of the agent and thereby improve the agent's performance.

Even though there is increased demand for online food delivery services, there is a huge concern about data privacy. Various mechanisms have been proposed to tackle the privacy issue, and there is a need to have an algorithm that can provide a tradeoff between utility and obfuscating the information.

**Author Contributions:** Conceptualization, Suleiman. Abahussein and D.Y.; methodology, C.Z., Z.C. and S.S.; software, S.A.; U.S.; validation, D.Y. and S.A.; formal analysis, S.A; investigation, S.A.; resources, S.A.; data curation, S.A.; writing—original draft preparation, S.A.; writing—review and editing, S.A. and D.Y.; visualization, S.A. and D.Y.; supervision, C.Z. and D.Y.; project administration, C.Z. and D.Y.; funding acquisition, C.Z. and D.Y. All authors have read and agreed to the published version of the manuscript.

**Funding:** This research received no external funding.

**Data Availability Statement:** We already mentioned the source of data and we cite the source in Datasets section.

**Conflicts of Interest:** The authors declare no conflict of interest.

**Abbreviations**

The following abbreviations are used in this manuscript:

| | |
|---|---|
| MARL | Multi-agent reinforcement learning |
| DRL | Deep reinforcement learning |
| DQN | Deep Q-network |
| IQL | Independent Q-learning |
| PULM | Protect User Location Method |
| DP | Differential privacy |

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
