# Peer review of "Multi-Agent Reinforcement Learning for Online Food Delivery with Location Privacy Preservation"

_information, doi:10.3390/info14110597_

Round 1

Reviewer 1 Report

Comments and Suggestions for Authors

1 The article is not highly related to agriculture.

2 The objectives of the research need to be clearly defined.

3 The abstract did not explain the results and draw conclusions.

4 The evaluation cannot be based solely on number of food delivery orders.

5 The algorithm can guide the couriers to the areas with a high demand for food delivery orders, which seems to imply a higher waiting time for the areas with a low demand for food delivery orders.

6 The article needs to clarify the purpose of privacy protection, which is to prevent who obtains private information.

7 There seems to be no significant correlation between privacy protection and area filtering. They are more inclined towards two independent topics.

8 The article does not propose a new algorithm, it is only an application of existing algorithms.

Comments on the Quality of English Language

Language is expected to be improved and polished.

Author Response

Comment 1: The article is not highly related to agriculture.

Response: We appreciate your comment, and we apologize for any confusion. Our research aligns with the journal's areas of interest, as our work focus on the application of artificial intelligence (specifically, multi-agent reinforcement learning) and data security. Our study addresses two critical challenges: improving the performance of online food delivery services and safeguarding user privacy within these platforms by preventing the disclosure of location information.

Comment 2: The objectives of the research need to be clearly defined.

Response: Thank you for your valuable comment. In this research we have two main objectives as the follow:

  • Introducing a method based on multi-agent reinforcement learning for online food delivery services, employing two multi-agent reinforcement learning algorithms. The principal aim of this approach is to increase the number of food delivery orders received, consequently enhancing the long-term income for couriers.

  • We propose a privacy-preserving defense method for safeguarding user privacy in online food delivery services. Our approach is designed to protect customer location information within online food delivery services using the differential privacy Laplace mechanism, achieved by inject Laplace noise to both customer locations and courier trajectories. The privacy parameter ε, which influences the level of noise injected, depends on two key factors: the size of the city area and the frequency of customer online food delivery orders.

Comment 3: The abstract did not explain the results and draw conclusions.

Response: Thank you for this insightful comment. We update the paper to comply with this comment.

Comment 4: The evaluation cannot be based solely on number of food delivery orders.

Response: Thank you for raising this issue. In this research, we propose two solutions: one for enhancing food delivery services to increase the number of orders and consequently boost long-term income. The results are based on

  • Multi agent reinforcement learning (MARL) rewards as the MARL get rewards when achieve the goal.
  • The average number of collected food delivery orders as this is the main objective.

The second solution was proposed a method to solve the privacy issue in online food delivery (location information) and we used multiple method to present our results:

  • Hausdorff Distance to evaluate the data utility and differences after adding Laplace
    noise and use our method to show the similarity between the two datasets of location (the original datasets and obfuscated datasets).
  • privacy parameter ε distribution generated by PULM algorithm

Comment 5: The algorithm can guide the couriers to the areas with a high demand for food delivery orders, which seems to imply a higher waiting time for the areas with a low demand for food delivery orders.

Response: Thank you for your comment. Directing couriers to high-demand areas can effectively reduce average waiting times. This approach ensures that a majority of couriers are dispatched to regions with a high demand for services. In contrast, sending couriers primarily to low-demand areas can result in extended waiting times for customers in high-demand areas, while couriers may find themselves with minimal tasks to perform.

Comment 6: The article needs to clarify the purpose of privacy protection, which is to prevent who obtains private information.

Response: We much appreciate your insightful comments. Thousands of food delivery orders are received by these food delivery services every single day, resulting in the collection of vast amounts of user data. This data could potentially be hosted by a third party and further processed for training and analysis. Additionally, the IT department might be outsourced to a third party. Various access permissions are granted to different types of data, creating opportunities for unauthorized access to customer information. Furthermore, adversaries could employ various attacks, such as inference attacks, to potentially infer sensitive information, thus posing a significant threat to customer privacy, including the disclosure of their location.

Comment 7: There seems to be no significant correlation between privacy protection and area filtering. They are more inclined towards two independent topics.

Response: We greatly appreciate your insightful feedback. Online food delivery services are essential to many people who rely on this solution for their meals. Our solution focuses on enhancing this service and we found there is potential threat that can come of using this service, leading us to propose a defence method.

Comment 8: The article does not propose a new algorithm, it is only an application of existing algorithms.

Response: Thank you for your comments. This research proposes two solution one for the online food delivery services and how increase the number of order and we propose PULM method to protect customer location information.

Reviewer 2 Report

Comments and Suggestions for Authors

Multi agent reinforcement learning for online food delivery with location privacy preserving

1. Very interesting research entitled “Multi agent reinforcement learning for online food delivery with location privacy preserving”.

2. Correct the structure of the article (see attached file).

** Check "Microsoft Word template" from information-MDPI.

https://www.mdpi.com/files/word-templates/information-template.dot

3. I suggest restructuring the article according to the journal template (link above).

4. Place the title of table 1 on a single line. Avoid using only capital letters. Use the format indicated below. (see attached file).

5. The equation of line 377 is not numbered. Correct.

6. In the middle of lines 431-438, is figure 2. The figure must be outside the paragraph.

7. In the middle of lines 570-573, there are figures 5 and 6. The figures must be outside the paragraph.

8. In the middle of lines 593-597, there are figures 7 and 8. The figures must be outside the paragraph.

9. The title of the figures (3, 4, 5, 6, 7 and 8) must be short. The previous paragraph should explain the figure.

10. The paragraph on lines 70-75 should be deleted, it is not appropriate to anticipate what will be dealt with in later sections.

11. How "Multi-agent reinforcement learning" is applied to train and guide the agent to the area with high demand for food delivery orders. Explain in detail.

12. How noise is injected for “preserving the privacy of customer location“, using PULM method. Explain the procedure in detail.

13. Consider future work on this research.

14. Very good bibliography.

The article has good content and very interesting.

Authors are requested to make all indicated corrections.

Author Response

Comment 1: Very interesting research entitled “Multi agent reinforcement learning for online food delivery with location privacy preserving”.

Response: Thank you for your feedback, and we appreciate this feedback.

Comment 2: Correct the structure of the article (see attached file).

Response: Thank you for your comments. We convert our paper to the structure you suggested but we add two more parts: Preliminary and Experiment Design

Comment 3: I suggest restructuring the article according to the journal template (link above).

Response: Thank you for your comments. We convert our paper to the structure you suggested but we add two more parts: Preliminary and Experiment Design

Comment 4: Place the title of table 1 on a single line. Avoid using only capital letters. Use the format indicated below. (see attached file).

Response: Thank you for your comment. We update the paper to comply with this comment.

Comment 5: The equation of line 377 is not numbered. Correct.

Response: Thank you for your comment. We update the paper to comply with this comment.

Comment 6: In the middle of lines 431-438, is figure 2. The figure must be outside the paragraph.

Response: Thank you for your comment. We update the paper to comply with this comment.

Comment 7: In the middle of lines 570-573, there are figures 5 and 6. The figures must be outside the paragraph.

Response: Thank you for your comment. We update the paper to comply with this comment.

Comment 8: In the middle of lines 593-597, there are figures 7 and 8. The figures must be outside the paragraph.

Response: Thank you for your comment. We update the paper to comply with this comment.

Comment 9: The title of the figures (3, 4, 5, 6, 7 and 8) must be short. The previous paragraph should explain the figure.

Response: Thank you for your comment. We update the paper to comply with this comment.

Comment 10: The paragraph on lines 70-75 should be deleted, it is not appropriate to anticipate what will be dealt with in later sections.

Response: Thank you for your feedback. We update the paper to comply with this comment.

Comment 11: How "Multi-agent reinforcement learning" is applied to train and guide the agent to the area with high demand for food delivery orders. Explain in detail.

Response: We employ multi-agent reinforcement learning to direct the courier to areas with high food delivery requests. We consider the city area as rectangular in shape, which we divide into N × M cells, each representing a small section of the city. Our multi-agent reinforcement learning approach employs a trial-and-error method for learning and exploration. Over time, the agent accumulates experience and refines itself. After training, the agent is capable of predict which areas with higher food delivery requests. Section 4.2 provides comprehensive details on this procedure.

Comment 12: How noise is injected for “preserving the privacy of customer location“, using PULM method. Explain the procedure in detail.

Response: Thank you for your feedback. We use Differential Privacy along with the Laplace mechanism by inject noise to random point of the courier trajectory and customer location we determined the privacy parameter (amount of noise) in this method based on two parameters. The first parameter is the city area size and the second parameter is the customer frequency of online food delivery orders. To calculate the privacy parameter, we sum the result from the city area size which has a weight of 2.5 with the result of the frequency of online food ordering which has a weight of 2.5, a smaller value of privacy parameter is stronger privacy. For example, the Iowa city area < 500 km2 this will gives 0.5 and let suppose we have a customer with 5 time of orders and in the dataset max(x) = 10 , min = 1. This will be calculated like (5-10) / (1-10) × 2.5 = 1.388, in this case, the privacy parameter will be 0.5 +1.388 = 1.88. In section 4.3 under Protect user location method PULM full details for the procedure.

Comment 13: Consider future work on this research.

Response: Thank you for your suggestion. We update the paper to comply with this comment.

Comment 14: Very good bibliography.

Response: Thank you for your feedback, and we appreciate this feedback.

Reviewer 3 Report

Comments and Suggestions for Authors

carefully check all comments and try to addressed all so that quality can be increase. 

Comments on the Quality of English Language

Minor required 

Author Response

Comment: The abstract is not written in good way. Carefully defined purposes, methods, and results.

Response: Thank you for your suggestion. We update the paper to comply with this comment.

Comment: Flow of study is missing at end of introduction.

Response: Thank you for your suggestion. We update the paper to comply with this comment.

Comment: Define list of abbreviations at the end of introduction and further used short term e.g. For machine learning used only ML after defining.

Response: Thank you for your suggestion. We update the paper to comply with this comment.

Comment: In literature there are two different style used for references. Some reference only number and some with name and number. correct it and use one style.

Response: Thank you for your feedback. We update the paper to comply with this comment.

Comment: Flow diagram of both algorithms missing. Draw it

Response: Thank you for your suggestion. We update the paper to comply with this comment.

Comment: Figures quality in results are very low.

Response: Thank you for your suggestion.

Comment: After equation 4 there is another equation but no number is assign.

Response: Thank you for your suggestion. We update the paper to comply with this comment.

Comment: Why author used this algorithm while some more better algorithm also available

Response: Thank you for your feedback. In this research, we propose two solutions: one for enhancing food delivery services to increase the number of orders and consequently boost long-term income. In this solution we use MARL as each courier consider as agent and these agents run together in same environment simultaneously. We use QMIX which is expected  to be based method acts in cooperation with nearby agents and IQL which consider as independent agent environment.

The second solution was proposed a method to solve the privacy issue in online food delivery (location information) and we used Differential privacy to protect the customer location information.

Comment: There is one diagram in introduction portion related to study and also one in literature review with high quality

Response: Thank you for your feedback.

Comment: Mentioned limitation of this algorithm

Response: Thank you for your suggestion. The main limitation of MARL is the limit number of agents that can work concurrently. The MARL (IQL) sometimes can’t converge in some complex environments. Also, the MARL come time could face  

Comment: There must be a table of comparison at the end of result in which this algorithm accuracy compares with some previous study.

Response: Based on our knowledge, the algorithms that we used along with datasets have not been applied to the same problem we tackle, so we may not be able to make a comparison with previous study.

Comment: There must be a table in literature review in which some previous study attributes (objectives, domains, advantages, limitation) are discussed.

Response: Thank you for your suggestion.

Comment: Carefully read following papers and some others and add a table in literature review.

Response: Thank you for your suggestion.

Round 2

Reviewer 1 Report

Comments and Suggestions for Authors

All my concerns have been well responded and revised. The currrent version of the manuscript is ready for consideration of publication.

Comments on the Quality of English Language

It's well written in terms of language.

Author Response

No comments from Reviewer 1

Reviewer 2 Report

Comments and Suggestions for Authors

I appreciate that you have made about 90% of the indicated observations.

Author Response

No comments from Reviewer 2

Reviewer 3 Report

Comments and Suggestions for Authors

I request you to carefully read all comments 

Comments on the Quality of English Language

Author Response

Comment :Carefully read the following papers and some others and add a table in the literature review. (This point is still unsolved.)

 Frikha, T., Ktari, J., Amor, N. B., Chaabane, F., Hamdi, M., Denguir, F., & Hamam, H. (2023). Low Power Blockchain in Industry 4.0 Case Study: Water Management in Tunisia. Journal of Signal Processing Systems, 1-15.

Balouch, S., Abrar, M., Abdul Muqeet, H., Shahzad, M., Jamil, H., Hamdi, M., ... & Hamam, H. (2022). Optimal scheduling of demand side load management of smart grid considering energy efficiency. Frontiers in Energy Research, 10, 861571.

Mazhar, T., Irfan, H. M., Haq, I., Ullah, I., Ashraf, M., Shloul, T. A., ... & Elkamchouchi, D. H. (2023). Analysis of Challenges and Solutions of IoT in Smart Grids Using AI and Machine Learning Techniques: A Review. Electronics, 12(1), 242.

Mazhar, T., Asif, R. N., Malik, M. A., Nadeem, M. A., Haq, I., Iqbal, M., ... & Ashraf, S. (2023). Electric Vehicle Charging System in the Smart Grid Using Different Machine Learning Methods. Sustainability, 15(3), 2603.

Mazhar, T., Malik, M. A., Mohsan, S. A. H., Li, Y., Haq, I., Ghorashi, S., ... & Mostafa, S. M. (2023). Quality of Service (quality of service) Performance Analysis in a Traffic Engineering Model for NextGeneration Wireless Sensor Networks. Symmetry, 15(2), 513.

Response: Thank you for your suggestion. Our research focuses on multi-agent reinforcement learning in online food delivery service and protecting the location information of the customer in online food delivery services.

We read through these papers mentioned above, and we not able to link these papers to our research. These papers focus on topics such as Water Management in Tunisia, smart grid, Electric Vehicle Charging Systems, Traffic Engineering models, IOT and Blockchain.

Instead we add the following paper to support literature review:

Adversarial attacks and defenses in deep learning: From a perspective of cybersecurity

Zhou, Shuai and Liu, Chi and Ye, Dayong and Zhu, Tianqing and Zhou, Wanlei and Yu, Philip S

Comment :The flow of the study is missing at the end of the introduction. Please carefully read the flow of study in which you mentioned what is in sections and sub-sections.

Response: Thank you for your feedback. The introduction section discusses two main issues:

  • The increase in the efficiency of online food delivery services and how to increase the number of food delivery orders and thereby increase long-term income.
  • How to protect customer information in online food delivery services.

Below is a summary of each segment of the introduction:

The first part offers an overview of the current state of online food delivery services, focusing on market value and people's usage habits. The second section highlights the challenges faced by the online food delivery services platforms, emphasizing potential factors that might reduce their efficiency. The third segment introduces our proposed solution for enhancing efficiency and augmenting the number of orders, ultimately aiming to increase long-term income.

The fourth section addresses the potential privacy concerns in online food delivery platforms. It sheds light on how adversaries might be able to access and reveal customer-specific data, such as location details. The fifth and final segment describes our defence method, explaining how it effectively safeguards customers' private information.     

Comment :After reading the abstract, I am still the same question where the authors have mentioned objectives, methods, results, and summary.

Response: Thank you for your feedback. This research has two main objectives: increase the number of received orders in the online food delivery services and thereby increase the long-term income of the courier. The second objective is protecting the customer location information in online food delivery services.

Our used method in this research is employ Multi-agent reinforcement learning to guide the couriers (agents) to the areas with a high demand for food delivery orders. The map of the city is divided into small grids, and each grid represents a small area of the city. The agent has to learn which grid has the highest demand for food delivery orders so it can select it. On the other hand, to protect customer location information, we invented the PULM method to protect customer location information by injecting Differential privacy Laplace noise based on two parameters: the city area size and the customer frequency of online food delivery orders.

We used two datasets, Shenzhen, China and Iowa, USA to demonstrate the result of our experiments. The result shows an increase in the number of received orders in the Shenzhen and Iowa datasets. We also show the similarity and data utility for courier trajectory after we use our obfuscation (PULM) method.

Comment : In literature, there are two different styles used for references. Some reference-only numbers, and some with names and numbers. Correct it and use one type. (This point is still unsolved.)

Response: Thank you for your feedback. We used the number as the main reference which provided by the journal latex templet. In the Literature Review, we wrote the author's name to distinguish between the different authors.

Comment :There is one diagram in the introduction portion related to the study and also one in the literature review with high quality (This point is still unsolved)

Response: Thank you for your feedback. I am sorry I am not able to get what you mean by this question.

Comment :There must be a table of comparison at the end of the result in which this algorithm's accuracy compares with some previous studies((This point is still unsolved)

Response: Thank you for your suggestion. We can’t make the comparison as based on our knowledge there are no any previous study applied their method to the same datasets (Shenzhen, China and Iowa) to solve same problems.

Comment :There must be a table in the literature review in which some previous study attributes (objectives, domains, advantages, and limitations) are discussed(((This point is still unsolved).

Response: Thank you for your suggestion. We update the paper by adding two table to comply with this comment.